# Water in the Alluaudite Type-Compounds: Synthesis, Crystal Structure and Magnetic Properties of $Co_3(AsO_4)_{0.5+x}(HAsO_4)_{2-x}$ $(H_2AsO_4)_{0.5+x}[(H,\square)_{0.5}(H_2O,H_3O)_{0.5}]^{2x+}$

Tamara Đorđević [1],*, Ljiljana Karanović [2], Marko Jagodič [3] and Zvonko Jagličić [3]

[1] Institut für Mineralogie und Kristallographie, Universität Wien-Geozentrum, Althanstrasse 14, 1090 Vienna, Austria
[2] Laboratory of Crystallography, Faculty of Mining and Geology, University of Belgrade, Đušina 7, 11000 Belgrade, Serbia; ljiljana.karanovic@rgf.bg.ac.rs
[3] Institute of Mathematics, Physics and Mechanics & Faculty of Civil and Geodetic, Engineering, University of Ljubljana, Jadranska 19, 1000 Ljubljana, Slovenia; marko.jagodic@imfm.si (M.J.); zvonko.jaglicic@imfm.si (Z.J.)
* Correspondence: tamara.djordjevic@univie.ac.at

**Abstract:** In this study, a new cobalt arsenate belonging to the alluaudite supergroup compounds with the general formula of $Co_3(AsO_4)_{0.5+x}(HAsO_4)_{2-x}(H_2AsO_4)_{0.5+x}[(H,\square)_{0.5}(H_2O,H_3O)_{0.5}]^{2x+}$ (denoted as CoAsAllu) was synthesized under hydrothermal conditions. Its crystal structure was determined by a room-temperature single-crystal X-ray diffraction analysis: space group $C2/c$, $a = 11.6978(8)$, $b = 12.5713(7)$, $c = 6.7705(5)$ Å, $\beta = 113.255(5)°$, $V = 914.76(11)$ Å$^3$, $Z = 2$ for $As_6H_8Co_6O_{25}$. It represents a new member of alluaudite-like protonated arsenates and the first alluaudite-like phase showing both protonation of the tetrahedral site and presence of the $H_2O$ molecules in the channels. In the asymmetric unit of CoAsAllu, one of the two Co, one of the two As and one of the seven O atoms lie at $4e$ special positions (site symmetry 2). The crystal structure consists of the infinite edge-shared $CoO_6$ octahedra chains, running parallel to the $[10\bar{1}]$ direction. The curved chains are interconnected by $[(As1O_4)_{0.5}(H_2As1O_4)_{0.5}]^{2-}$ and $[HAs_2O_4]^{2-}$ tetrahedra forming a heteropolyhedral 3D open framework with two types of parallel channels. Both channels run along the $c$-axis and are located at the positions $(1/2, 0, z)$ and $(0, 0, z)$, respectively. The H2 and H4 hydrogen atoms of O2H2 and O4H4 hydroxyl groups are situated in channel 1, while the uncoordinated water molecule $H_2O7$ at half-occupied 4e special positions and hydrogen atoms of O6H6 hydroxyl group were found in channel 2. The results of the magnetic investigations confirm the quasi one-dimensional structure of divalent cobalt ions. They are antiferromagnetically coupled with the intrachain interaction parameter of $J \approx -8$ cm$^{-1}$ and interchain parameter of $J' \approx -2$ cm$^{-1}$ that become effective below the Néel temperature of 3.4 K.

**Keywords:** alluaudite supergroup; transition metal arsenates; hydrothermal synthesis; crystal structure; hydrated hydroxy-arsenates; stoichiometry; magnetic properties

## 1. Introduction

In the past two decades, structural crystallography of the synthetic analogues of the minerals adopting alluaudite-type structure have attracted great scientific attention because of their open-framework structure and their unique physical properties [1,2], which raise interest in their potential applications e.g., corrosion inhibition, metal surfaces passivation, energy storage and catalysis [3–7]. Advances in synthesis techniques have allowed experts to substitute a wide range of elements into the particularly flexible open-framework of the alluaudite-type structure [8].

Minerals of the alluaudite-group are complex primary and secondary phosphates and arsenates that occur primarily in granitic pegmatites, but have also been found in scoria,

phosphatic nodules, granites and volcanic fumaroles. There are currently 25 minerals (6 phosphates and 19 arsenates) [9]. The alluaudite supergroup minerals are further divided into the following four groups: alluaudite-, wyllieite-, bobfergusonite- and manitobaite-groups [10]. The minerals of these groups share the same topology but are differentiated by diverse cation-ordering schemes over the octahedrally coordinated $M$-sites. They are all monoclinic (space group $C2/c$).

The general formula of the alluaudite-type compounds can be written as $[A2A2'A2''_2]$ $[A1A1'A1''_2]M1M2_2(TO_4)_3$ [11] or $A1A2M1M2_2(T1O_4)(T2O_3OH)_2$ for the protonated members of the supergroup [12]. Minerals belonging to the alluaudite-group compounds are usually phosphates and arsenates, with Na, Ca, Mn occupying $A$-positions and Al, Mg, Mn and Fe occupying $M$-positions. Recently, the nomenclature of the group has been revised [10]. The simplified structural formula for the alluaudite-type compounds is now $A2'A1M1M2_2(TO_4)_3$ for Z = 4. Synthetic members of the group exhibit much larger chemical variability. In these compounds, distinctly different cation sites for $A1$ and $A2$ are found, i.e., the monovalent cations may be replaced by protons leading to the compounds $AM_3(XO_4)(XO_3OH)_2$, where $A$ = $H^+$, $Na^+$, $K^+$, $Ag^+$ and $Cd^{2+}$ [13–16]. The structural and chemical flexibility of this structure type reflects in the fact that mineral species belonging to the alluaudite structure-type are yet to be discovered Many of these new discoveries, such as minerals johillerite, $NaCuMg_3(AsO_4)_3$, badalovite, $NaNaMg(MgFe^{3+})(AsO_4)_3$ [17], camanchacaite, $NaCaMg_2[AsO_4]_2[AsO_3(OH)_2]$ [18] and calciojohillerite, $NaCaMgMg_2(AsO_4)_3$ [19] are arsenates.

The focus on new arsenates and arsenites in the systems that are of magnetic significance has led us to perform a series of low-temperature hydrothermal experiments in the $A_2O–MO–As_2O_5(As_2O_3)–H_2O$ and $AO–MO–As_2O_5(As_2O_3)–H_2O$ systems ($A$ = $Na^+$, $K^+$, $Sr^{2+}$, $Ba^{2+}$, $Cd^{2+}$; $M$ = $Mg^{2+}$, $Mn^{2+,3+}$, $Fe^{2+,3+}$, $Co^{2+}$, $Ni^{2+}$, $Cu^{2+}$, $Zn^{2+}$). Accordingly, many new $A–M–$ and $A–M–$(H–) arsenates and arsenites were characterized structurally and spectroscopically [12,20–22]. Among the arsenates, alluaudite-type compounds are of particular interest because they are formed as the result of the interactions of arsenate anions with transition metal and alkali metal cations. The incorporation of transition elements is particularly attractive due to the opportunity to identify interesting magnetic properties [23,24].

In the present article, we report on the hydrothermal synthesis, crystal structure and on selected magnetic properties of $Co_3(AsO_4)_{0.5+x}(HAsO_4)_{2-x}(H_2AsO_4)_{0.5+x}[(H,\square)_{0.5}(H_2O, H_3O)_{0.5}]^{2x+}$ (denoted as CoAsAllu) with alluaudite-type crystal structure, the first alluaudite-type compound showing both protonation of the tetrahedral site and presence of the water molecules in the channels.

## 2. Materials and Methods

Single crystals of CoAsAllu were synthesized hydrothermally under autogenous pressure from a mixture of 0.2 g of $Co(NO_3)_2 \cdot 6H_2O$ (Sigma–Aldrich; 239267-100G; 98%)·and 0.2 g of $As_2O_5$ (Alfa Aesar 14668, 100 g, 99.9%) and 0.4. ml of distilled water. The Teflon vessel was filled with this mixture (pH =5) and distilled water to approximately 75% of its inner volume and was enclosed in a stainless steel autoclave. The following three-step heating regime was chosen: the autoclave was heated from 293 to 493 K (4 h), held at 493 K for 48 h and finally cooled to room temperature within 156 h. The pH of the remaining solution was 2. The reaction products were filtered off and washed carefully with distilled water. CoAsAllu was crystallized as transparent dark-pink rod-like crystals up to 0.2 mm in length.

The CoAsAllu was further characterized using room-temperature single-crystal X-ray diffraction (SXRD) (Enraf-Nonius Corporation, Delft, Netherlands), scanning electron microscopy with energy dispersive spectroscopy (SEM-EDS) (JEOL Ltd., Akishima, Tokyo, Japan), Fourier transformed infrared (FTIR) (Bruker-AXS, Madison, WI, USA) and Raman spectroscopy (K.K. Horiba Seisakusho, Kyoto, Japan). Its magnetic properties were determined by means of the magnetic susceptibility measurements.

### 2.1. Single Crystal X-ray Diffraction Analysis

The quality of a few single crystals of CoAsAllu was checked with a STOE StadiVari single-crystal four-circle diffractometer (Mo tube, plane graphite monochromator, pixel array Pilatus 300K detector) (Enraf-Nonius Corporation, Delft, Netherlands). The most suitable crystal was chosen for data collection. A complete sphere of reciprocal space ($\varphi$ and $\omega$ scans) was measured at room temperature. The intensity data were processed with the STOE program suite *X-AREA* and *X-RED32* [25], corrected for Lorentz, polarization, and background effects and, by the multi-scan method [25], for absorption. The crystal structure was solved by direct methods using *SIR2014* [26] and refined on $F^2$ by full-matrix least-squares using *SHELXL97* [27] and *WinGX* [28]. Initially the Co, As and O1-O6 sites were only refined. As expected, the refinements of these sites imply that they all were fully occupied. Then, in special positions 4*e* (site symmetry is 2) in channel 2, there appears a low-density peak which corresponds to half of an oxygen, i.e., the s.o.f. of this oxygen, O7, had to be reduced to 0.5 (50% occupancy), which means that only two O7 atoms exist per unit cell.

Since the H atoms of hydroxyl O2H2, O4H4 and O6H6 groups could not be found in a difference Fourier map, their positions were positioned geometrically and refined using a riding model HFIX 83 with $U_{iso}(H) = 1.2\ U_{eq}(O)$, distance restraints for O–H = 0.82(2) Å and with $M$–O–H ($M$ = As, Co) angle tetrahedral. Two hydrogen atoms of water molecule H2O7 inside the channel have not been located because of disorder and their small amount. Anisotropic displacement parameters for all non-hydrogen atoms were enabled to vary. The anisotropies were found to be only moderate except for water oxygen O7, probably due to its partial occupancy and the possible rotational and translational motion. Although it was only the hydrogen atoms of this hydrogen bonded water molecule that were not located, it is obvious that they form hydrogen bonds with adjacent oxygen O6 from hydroxyl O6H6 group situated in channel 2 (O6$\cdots$O7$^{ix}$ = 2.261(2) Å; symmetry code: (ix) $-x + 1$, $-y + 1$, $-z + 1$). The least-squares refinement led to the final reliability factors ($R[F^2 > 2\sigma(F^2)]$= 0.020 $wR(F^2)$ = 0.043), obtained by fitting 88 parameters. The final difference Fourier map had maximum and minimum peaks of 0.67 and $-1.49$ eÅ$^{-3}$.

Crystal data, information on the data collection, and results of the final structure refinement are compiled in Table 1. The fractional atomic coordinates, occupancies, and equivalent or isotropic atomic displacement parameters are given in Table 2 and anisotropic atomic displacement parameters in Table 3. Selected bond distances and bond angles as well as bond valence sum calculations obtained using *VALIST* software (A. S. Wills, UCL Berkeley, USA) [29] are presented in Table 4. The hydrogen bonds are presented in Table 5. All drawings of structure were produced with *ATOMS* (Shape Software, Kingsport, TN, USA) [30]. CDS 2117677 contains the supplementary crystallographic data for this paper. The data can be obtained free of charge via www.ccdc.cam.ac.uk/data_request/cif, accessed on 25 October 2021, or by emailing data_request@ccdc.cam.ac.uk, or by contacting The Cambridge Crystallographic Data Centre, 12, Union Road, Cambridge CB2 1EZ, UK, fax: +44 1223 336033.

**Table 1.** Experimental data.

| Crystal Data | |
| --- | --- |
| Chemical formula | As6H8Co6O25 |
| Molecular mass, *Mr* | 1211.16 |
| Crystal system, space group | Monoclinic, *C2/c* |
| Temperature (K) | 293 |
| *a, b, c* (Å) | 11.6978(8), 12.5713(7), 6.7705(5) |
| $\beta$ (°) | 113.255(5) |
| *V* (Å3) | 914.76 (11) |
| Z | 2 |
| Radiation type | Mo *Kα* |
| $\mu$ (mm$^{-1}$) | 16.222 |
| Crystal size (mm) | 0.08 × 0.02 × 0.02 |

**Table 1.** *Cont.*

| Data Collection | |
|---|---|
| Diffractometer | STOE StadiVari |
| Absorption correction | Multi-scan |
| No. of measured, independent and observed [$I > 2\sigma(I)$] reflections | 8740, 1639, 1414 |
| $R_{int}$ | 0.041 |
| $(\sin\theta/\lambda)_{max}$ (Å$^{-1}$) | 0.757 |
| **Refinement** | |
| $R[F^2 > 2\sigma(F^2)]$, $wR(F^2)$, $S$ | 0.020, 0.043, 1.00 |
| No. of reflections | 1639 |
| No. of parameters | 88 |
| H-atom treatment | H-atom parameters constrained |
| $\Delta\rho_{max}$, $\Delta\rho_{min}$ (eÅ$^{-3}$) | 0.67, −1.49 |

**Table 2.** Fractional atomic coordinates and isotropic or equivalent displacement parameters (Å$^2$).

| Atom | $x$ | $y$ | $z$ | $U_{iso}$*/$U_{eq}$ | Occupancy (<1) |
|---|---|---|---|---|---|
| As1 | 0.500000 | 0.81531(2) | 1.250000 | 0.00713(6) | |
| As2 | 0.29008(2) | 0.61716(2) | 0.39402(3) | 0.00896(6) | |
| Co1 | 0.500000 | 0.78694(3) | 0.750000 | 0.00777(8) | |
| Co2 | 0.30090(3) | 0.66037(2) | 0.89043(4) | 0.00849(6) | |
| O1 | 0.46638(14) | 0.74458(12) | 1.0241(2) | 0.0084(2) | |
| O2 | 0.62013(14) | 0.89752(13) | 1.2730(3) | 0.0111(3) | |
| H2 | 0.604032 | 0.958135 | 1.298744 | 0.013* | 0.5 |
| O3 | 0.35017(15) | 0.68027(13) | 0.6308(2) | 0.0098(3) | |
| O4 | 0.13697(15) | 0.59005(13) | 0.3394(3) | 0.0122(3) | |
| H4 | 0.132130 | 0.554871 | 0.437751 | 0.015* | 0.5 |
| O5 | 0.29295(15) | 0.68858(12) | 0.1877(2) | 0.0093(3) | |
| O6 | 0.36646(16) | 0.49889(13) | 0.4159(3) | 0.0149(3) | |
| H6 | 0.439143 | 0.510404 | 0.435550 | 0.018* | 0.5 |
| O7 | 0.500000 | 0.5129(8) | 0.750000 | 0.048(2) | 0.5 |

**Table 3.** Atomic displacement parameters (Å$^2$).

| Atom | $U^{11}$ | $U^{22}$ | $U^{33}$ | $U^{12}$ | $U^{13}$ | $U^{23}$ |
|---|---|---|---|---|---|---|
| As1 | 0.00900(13) | 0.00658(12) | 0.00501(11) | 0.000 | 0.00190(9) | 0.000 |
| As2 | 0.01160(10) | 0.00900(10) | 0.00620(8) | −0.00325(7) | 0.00345(7) | −0.00085 (6) |
| Co1 | 0.00725(16) | 0.00874(16) | 0.00791(15) | 0.000 | 0.00362(12) | 0.000 |
| Co2 | 0.00752(13) | 0.01095(13) | 0.00756(11) | −0.00178(9) | 0.00357(9) | −0.00083(8) |
| O1 | 0.0077(6) | 0.0119(6) | 0.0057(5) | −0.0030(5) | 0.0027(4) | −0.0024(5) |
| O2 | 0.0078(6) | 0.0082(6) | 0.0172(6) | −0.0007(5) | 0.0048(5) | −0.0001(5) |
| O3 | 0.0105(6) | 0.0129(7) | 0.0063(5) | −0.0040(5) | 0.0036(5) | −0.0023(5) |
| O4 | 0.0088(6) | 0.0095(6) | 0.0185(7) | −0.0007(5) | 0.0057(5) | −0.0002(5) |
| O5 | 0.0107(6) | 0.0114(6) | 0.0069(5) | 0.0007(5) | 0.0045(5) | 0.0018(5) |
| O6 | 0.0131(7) | 0.0075(6) | 0.0241(8) | 0.0000(5) | 0.0073(6) | −0.0019(6) |
| O7 | 0.036(4) | 0.076(6) | 0.028(3) | 0.000 | 0.010(3) | 0.000 |

*2.2. Chemical Composition and Micromorphology*

Qualitative chemical analyses were performed using a JEOL JSM-6610 LV scanning electron microscope (SEM) with W-filament (15 kV, working distance 17 mm), equipped with SE- and BSE-detectors and EBSD and cathodoluminescence units. The investigated single crystals (Figure 1) were fixed with conducting carbon onto Al-sample holders and were consequently coated with a thin layer of carbon for chemical analysis. EDS analysis proved the presence of each reported element in the CoAsAllu.

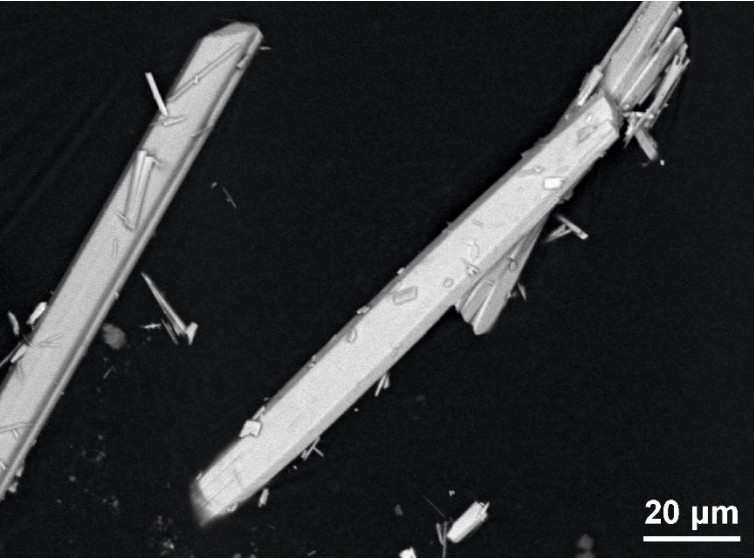

**Figure 1.** The back-scattered electron image of the idiomorphic crystals of $Co_3(AsO_4)_{0.5+x}$ $(HAsO_4)_{2-x}(H_2AsO_4)_{0.5+x}[(H,\square)_{0.5}(H_2O,H_3O)_{0.5}]^{2x+}$.

Semi-quantitative chemical analysis revealed the presence of cobalt in the range of 15.5–17.4 at%, arsenic in the range of 16.4–19.6 at%. Atomic proportions of main elements calculated from atomic %, Co:As = 1.05 correspond well with the ratio of Co:As =1 obtained from the structural analyses.

### 2.3. Vibrational Spectroscopy

In order to investigate the hydrous species in more detail, Fourier-transform infrared (FTIR) absorption single-crystal infrared spectra were recorded on a Bruker Tensor 27 FTIR spectrophotometer with a mid-IR glowbar light source and KBr beam splitter, attached to a Hyperion 2000 FTIR microscope with liquid nitrogen-300 cooled mid-IR broad band MCT detector. For the measurement, the sample was prepared as a KBr pellet. A total of 128 scans were accumulated between 4000 and 400 $cm^{-1}$ using the sample prepared as KBr micro-pellet and a total of 128 scans were accumulated between 4000 and 400 $cm^{-1}$ using a single crystal, circular sample aperture 100 µm diameter and ATR 15× objective.

For the detailed study of the arsenate groups, single-crystal Raman spectra of CoAsAllu was measured with a Horiba LabRam–HR system equipped with Olympus BX41 optical microscope in the spectral range between 100 and 4000 $cm^{-1}$. The 632.8 nm excitation line of a He–Ne laser was focused with a 100× objective (N.A. = 0.90) on the randomly oriented single crystal. The nominal exposure time of the aquired spectry was 10 s (confocal mode, Olympus 1800 lines/mm, 1.5 µm lateral resolutions, and approximately 3 µm depth resolution). The density of the laser power was well below the threshold for possible sample changes due to intense laser-light absorption and resulting temperature increase.

### 2.4. Magnetic Properties

Magnetic properties of CoAsAllu were examined on polynuclear sample using a Quantum Design MPMS-XL-5 SQUID magnetometer (San Diego, CA 92121, USA). Susceptibility has been measured between 2 K and 300 K in a constant magnetic field of 500 Oe, and the magnetization was measured between −50 kOe and 50 kOe at different temperatures. The temperature dependent susceptibility was measured following two protocols: heating the sample in magnetic field after the sample has been cooled in zero field (zfc susceptibility), and cooling the sample in magnetic field (field-cooled, fc susceptibility). All data were corrected for temperature independent contribution of core electrons as obtained from Pascall's tables [31].

## 3. Results and Discussion

### 3.1. Description of the Crystal Structure

The crystal structure of CoAsAllu (Figure 2) is comparable to the other protonated arsenates of the alluaudite supergroup. The main characteristic of the CoAsAllu crystal structure is the infinite edge-shared $CoO_6$ octahedra chain, running parallel to the [10$\bar{1}$] direction of the monoclinic unit cell. The fractional atomic coordinates, occupancies and equivalent or isotropic atomic displacement parameters are listed in Table 2. The chains are interconnected by $[(As1O_4)_{0.5}(H_2As1O_4)_{0.5}]^{2-}$ and $[HAs2O_4]^{2-}$ tetrahedra forming heteropolyhedral 3D open framework with two types of parallel channels. Both channels 1 and 2 run along the *c*-axis and are located at $(1/2, 0, z)$ and $(0, 0, z)$, respectively (Figure 2d).

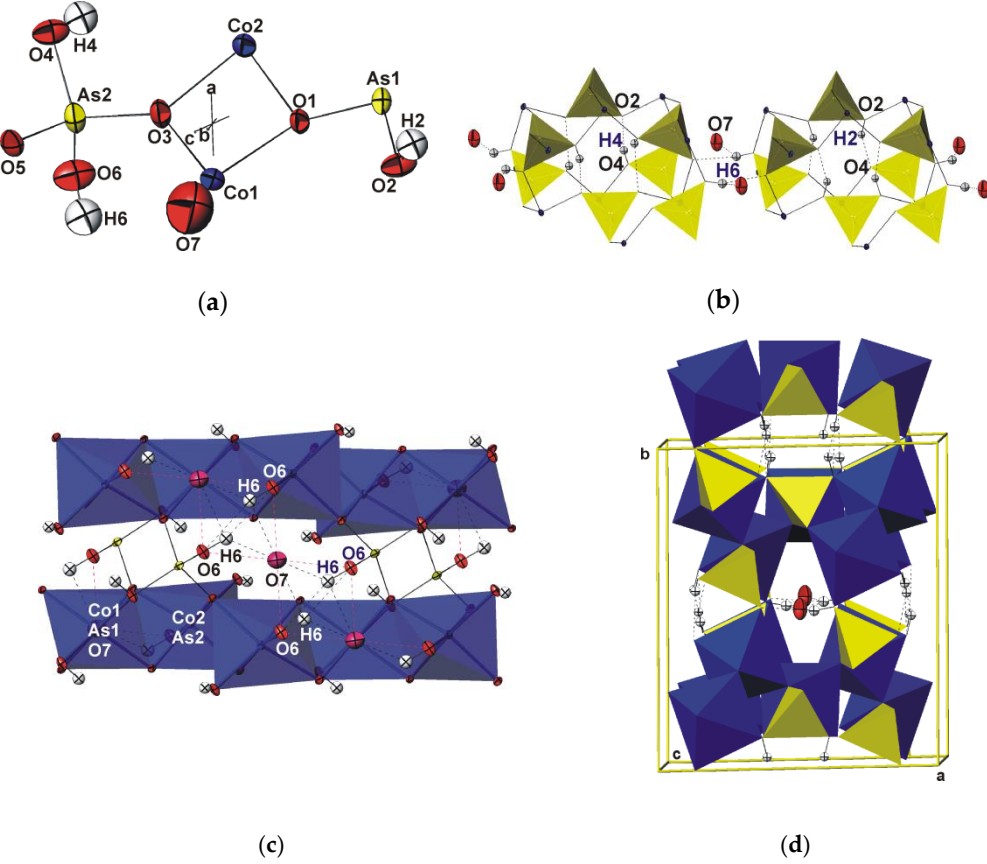

**Figure 2.** The asymmetric unit with the atom-numbering scheme (**a**); hydrogen bonds within the channel 1 and bifurcated hydrogen bonds O6—H6···O6$^{ix}$ and O6—H6···O7 $^{ix}$ (symmetry code: (ix) $-x + 1, -y + 1, -z + 1$) in channel 2 (**b**); the view along [010] of curved chains composed of edge-shared Co1O$_6$ and Co2O$_6$ octahedra with water oxygen O7 acting as single hydrogen bond acceptor of H6 (black dotted lines) and as a possible double bond donor toward four symmetry-equivalents of adjacent O6 (red dotted lines) (**c**); the projection of the structure nearly along [001] direction (**d**). Displacement ellipsoids are plotted at the 90% probability level. The octahedral chains (blue) with arsenate tetrahedra (greenish-yellow), H atoms of hydroxyl groups (light gray circles) in both channels and O7 atoms (red ellipsoids) of water molecule in channel 2. In both channels, the half of H and O7 atoms is arbitrary omitted in order to illustrate their half-occupied positions.

As mentioned, the structure contained situated hydrogen atoms of hydroxyl O2H2 and O4H4 groups in channel 1, while the uncoordinated water molecules $H_2O7$ at half-occupied 4*e* special positions, along with hydroxyl O6H6, were found in channel 2, which means that only two water molecules exist per unit cell.

The Co1 and As1 cations and O7 oxygen atoms from water molecules are at two-fold rotation axes in special 4*e* positions, while Co2, As2 and O1–O6 oxygen atoms are in general 8*f* positions (Figure 2a). The curved chains are built of slightly distorted Co1O$_6$ and Co2O$_6$ octahedra, adopting the bond distance scheme 4 + 2 (Figure 2c,d). Each CoO$_6$ octahedra

shares two edges with two adjacent octahedra and all six corners with adjacent arsenate tetrahedra. The individual Co1–O bond lengths vary from 2.0987(15) to 2.1352(17) Å, while Co2–O bond lengths vary from 2.0702(15) to 2.1523(15) Å (Table 4). The average <Co(II)–O> bond distances are similar: <2.115> and <2.097> Å for <Co1–O> and <Co2–O>. These average values are close to those calculated from effective ionic radii of six-coordinated Co(II) ions (0.745 + 1.38 = 2.125 Å), [32]. The bond angles around the central metal ions Co1 and Co2 deviate somewhat from the ideal octahedral molecular geometry [*cis* angles: 76.11(6)–108.24(6)° and *trans* angles: 150.78(9)–172.32(6)° for Co1; *cis* angles: 77.53(6)–107.48(6)° and *trans* angles: 159.10(6)–169.49(6)° for Co2]. The cobalt$\cdots$cobalt distances in chains are 3.1769(2) and 3.2571(2) Å for Co2$\cdots$Co2$^x$ (symmetry code: (x) $-x + 1/2$, $-y + 1/2$, $-z + 1$) and Co2$\cdots$Co1, respectively, which are about 1.5 Å shorter than the sum of van der Waals radii (2 × 2.4 Å) [33], indicating van der Waals interactions between Co atom pairs. The adjacent CoO$_6$ octahedra chains are linked together by the AsO$_4$ tetrahedra corners.

**Table 4.** Bond lengths (Å), bond-valences and bond valence sums (v.u.).

| Atom1-Atom2 | Bond lengths (Å) | Bond Valences (v.u.) | Atom1-Atom2 | Bond Lengths (Å) | Bond Valences (v.u.) |
|---|---|---|---|---|---|
| As1—O1 | 1.6763(14) | 1.279 | As2—O5 | 1.6718(15) | 1.293 |
| As1—O1$^i$ | 1.6763(14) | 1.279 | As2—O3 | 1.6742(15) | 1.286 |
| As1—O2 | 1.7012(16) | 1.195 | As2—O6 | 1.7106(16) | 1.163 |
| As1—O2$^i$ | 1.7012(16) | 1.195 | As2—O4 | 1.7139(16) | 1.154 |
| Average/$\Sigma\nu_{ij}$ | <1.689> | 4.948(1) | | <1.693> | 4.896(2) |
| Co1—O3 | 2.0987(15) | 0.333 | Co2—O3 | 2.0702(15) | 0.360 |
| Co1—O3$^{iii}$ | 2.0987(15) | 0.333 | Co2—O1 | 2.0745(15) | 0.355 |
| Co1—O1$^{iii}$ | 2.1114(14) | 0.322 | Co2—O2$^{vi}$ | 2.0750(16) | 0.355 |
| Co1—O1 | 2.1115(14) | 0.322 | Co2—O5$^{ii}$ | 2.0818(15) | 0.349 |
| Co1—O4$^{vii}$ | 2.1352(17) | 0.302 | Co2—O6$^v$ | 2.1265(16) | 0.309 |
| Co1—O4$^{iv}$ | 2.1352(17) | 0.302 | Co2—O5$^{iv}$ | 2.1523(15) | 0.288 |
| average/$\Sigma\nu_{ij}$ | <2.115> | 1.914 (4) | | <2.097> | 2.016(1) |

Symmetry codes: (i) $-x + 1$, $y$, $-z + 5/2$; (ii) $x$, $y$, $z + 1$; (iii) $-x + 1$, $y$, $-z + 3/2$; (iv) $-x + 1/2$, $-y + 3/2$, $-z + 1$; (v) $x$, $-y + 1$, $z + 1/2$; (vi) $x - 1/2$, $-y + 3/2$, $z - 1/2$; (vii) $x + 1/2$, $-y + 3/2$, $z + 1/2$.

Arsenate anions are present as diprotonated (H$_2$As1O$_4$)$^-$, monoprotonated (HAs2O$_4$)$^{2-}$ and deprotonated (As1O$_4$)$^{3-}$ species. Although the mean <As–O> bond lengths are almost the same for both arsenate tetrahedra (<As1–O> = <1.689>, <As2–O> = <1.693> Å), the individual bond lengths vary from 1.6763(14) to 1.7012(16) and from 1.6718(15) to 1.7139(16) Å for As1–O and As2–O, respectively. These average distances are somewhat shorter than the sum of the effective ionic radii of As(V) in tetrahedral coordination and O$^{2-}$ (0.335 + 1.38 = 1.715 Å) [32].

The arsenic As1(V) cation, located at the 4*e* special position, is coordinated by four adjacent oxygens, two symmetry-equivalents of O1 and two of O2, adopting a tetrahedral coordination. There are slight deviations from the ideal 109.5° angle in a regular tetrahedron: the O–As1–O bond angles deviate from 105.19(11) to 115.92(11)°. The two O2 are at somewhat longer distances from the arsenic compared to the other two, and this suggests that O2 is protonated to a hydroxyl group. Owing to the symmetry requests, As1O$_4$ tetrahedra are either deprotonated or diprotonated (both symmetry-equivalents of O2). Therefore, the oxygens O2 act as hydrogen bond donors to the oxygens O4 in one-half and as single hydrogen bond acceptors from O4H4 groups in the second half of unit cells (Figure 2b).

Likewise, the oxygens O4 from previously mentioned O4H4 groups act as hydrogen bond donors to the oxygens O2, in one half, and as single hydrogen bond acceptors from O2H2, in the second half time. Both H2 and H4 hydrogen atoms are located at half-occupied 8*f* general positions, so that only half of the sites are occupied by OH and another half are occupied by O. The H2 and H4 atoms are statistically distributed among their two

positions within channel 1 and are involved in moderately strong hydrogen bonds ($D \cdots A$ distances O2—H2$\cdots$O4$^{viii}$ and O4—H4$\cdots$O2$^{vi}$ [symmetry codes: (viii) $x + 1/2, y + 1/2, z + 1$] (vi) $x − 1/2, −y + 3/2, z − 1/2$] are 2.455(2) and 3.023(2) Å, respectively (Table 5).

**Table 5.** Hydrogen-bond geometry with H$\cdots A < r_A$ + 2 Å ($r_A$ is a radius of an acceptor) and $D$–H$\cdots A$ > 110°.

| $D$—H$\cdots A$ | $D$—H | H$\cdots A$ | $D \cdots A$ | $D$—H$\cdots A$ |
|---|---|---|---|---|
| O2—H2$\cdots$O4$^{viii}$ | 0.82 | 1.70 | 2.455(2) | 152 |
| O4—H4$\cdots$O2$^{vi}$ | 0.82 | 2.40 | 3.023(2) | 133 |
| O6—H6$\cdots$O6$^{ix}$ | 0.82 | 2.10 | 2.872(4) | 158 |
| O6—H6$\cdots$O7$^{ix}$ | 0.82 | 1.70 | 2.261(2) | 124 |

Symmetry codes: (vi) x − 1/2, −y + 3/2, z − 1/2; (viii) x + 1/2, y + 1/2, z + 1; (ix) −x + 1, −y + 1, −z + 1.

Another arsenic cation, As2(V), is located at the *8f* general position. From the As2—O distances, it is clear that the O4 and O6 oxygens bonded to As2 are protonated. Likely, the As—OH bond distances are slightly extended in comparison to the other As2—O bonds in tetrahedra (Table 4). The O–As2–O bond angles are within the interval of 106.88(8)–113.96(7)°. The O6H6 group acts as a single hydrogen-bond donor to the symmetry-equivalent oxygen O6$^{ix}$ (symmetry code: (ix) $−x + 1, −y + 1, −z + 1$). The O6–H6$\cdots$O6$^{iii}$ is a somewhat bent hydrogen bond ($D \cdots A$ distance is 2.872(4) Å and $D$–H$\cdots A$ angle is 158°). The H6 atom is located in channel 2 on the half-occupied *8f* general position similar to H2 and H4, so that only half of the sites are occupied by O6H6 and the additional half are occupied by O6.

The O6H6 group also acts as a single hydrogen-bond donor to the oxygen from a water molecule, O7, making the second considerably bent hydrogen bond ($D \cdots A$ distance is 2.261(2) Å and $D$–H$\cdots A$ angle is 124°). Therefore, H6 participates in two hydrogen bonds: O6–H6$\cdots$O6$^{ix}$ and O6–H6$\cdots$O7$^{ix}$. Besides as a single hydrogen bond acceptor of H6, the oxygen O7 additionally acts as a double bond donor to possibly four symmetry-equivalents of adjacent O6 (Figure 2c). Bond-valence calculations [34] demonstrate that the Co–O and As–O bond lengths are consistent with the presence of Co(II), As(V) and O$^{2−}$ ($\Sigma v_{ij}$ = 1.914(4) v.u. for Co1, $\Sigma v_{ij}$ = 2.016(1) v.u. for Co2 with CN = 6, $\Sigma v_{ij}$ = 4.948(1) v.u. and $\Sigma v_{ij}$ = 4.896(2) v.u. for As1 and As2, respectively, with CN = 4). Taking into consideration that only the contribution of the non-hydrogen atoms, the oxygens involved in hydrogen bonding, are expectedly undersaturated: $\Sigma v_{ij}$ are 1.55(2), 1.46(3), 1.48(3) v.u. for O2, O4 and O6, respectively. Concerning the fact that O2, O4, and O6 are alternately single hydrogen bond donors and acceptors to one another, bond valence values are very well balanced. The bond valence sums for other oxygens are near to the nominal valence of 2–: $\Sigma v_{ij}$ are 1.955(2), 1.979(1) and 1.930(4) v.u. for O1, O3 and O5, respectively.

### 3.2. Infrared and Raman Spectra of CoAsAllu

For the purpose of studying the hydroxyl group content in the CoAsAllu, FTIR spectrum was recorded. The existence of hydrogen bonds can be determined by infrared spectroscopy in the regions of the OH stretching modes. Owing to the strong absorption, bands under 1000 cm$^{-1}$ could not be analyzed. Hence, the diverse frequency regions may be allocated as follows: the spectral range between 3500 and 1000 cm$^{-1}$ shows a rare escalation in its "background absorption" (Figure 3a), which is a typical feature of the compounds with very short hydrogen bonds, but also is the key spectral feature of structurally similar synthetic protonated alluaudite-like arsenates [12,13]. As–O tetrahedral stretching vibrations and lattice modes are superimposed in the low-energy region of the spectra due to the shape of the broadband. The powder infrared spectrum clearly shows the presence of broad O–H stretching vibrations centered around 3200/2935/2382 cm$^{-1}$. These values compare well with the O–H stretching frequencies of the protonated AsO$_4$ groups in Cd$_{1.16}$Zn$_{0.34}$(AsO$_4$)$_{1.5}$(HAsO$_4$)(H$_2$AsO$_4$)$_{0.5}$ (3236 and

2300 cm$^{-1}$), Cd$_{0.74}$Mg$_{2.76}$(AsO$_4$)$_{1.5}$(HAsO$_4$)(H$_2$AsO$_4$)$_{0.5}$ (3210 and 2390 cm$^{-1}$) [13] and NaMg$_3$(AsO$_4$)(AsO$_3$OH)$_2$, NaZn$_3$(AsO$_4$)(AsO$_3$OH)$_2$ (2900/2300 cm$^{-1}$) [15] and Cd$_{0.75}$ Co$_{2.75}$(H$_{0.5}$AsO$_4$)$_2$(HAsO$_4$) [21]. The ratio of the bands at 3412 and 815 cm$^{-1}$ corresponds to high-hydrous acid arsenates with H$_2$O: As > 3 [35]. Thus, one could think that the bands at 3412 and 1641 cm$^{-1}$ are mainly due to water molecules adsorbed by the KBr pallet. However, the IR-spectrum measured using the ATR-technique unambiguously showed the presence of the water molecule (Figure 3a) (3344 and 1626 cm$^{-1}$). In the ATR-spectrum of CoAsAllu, the bands are slightly shifted to lower wave numbers caused by dispersion of the refractive index. The bands centered around ~1237 $\pm$ 20 cm$^{-1}$ may correspond to the vibrations of the free H$^+$ cation formed as the result of the dissociation of the acid arsenate groups [35,36], but also can represent the As–O–H bending vibrations.

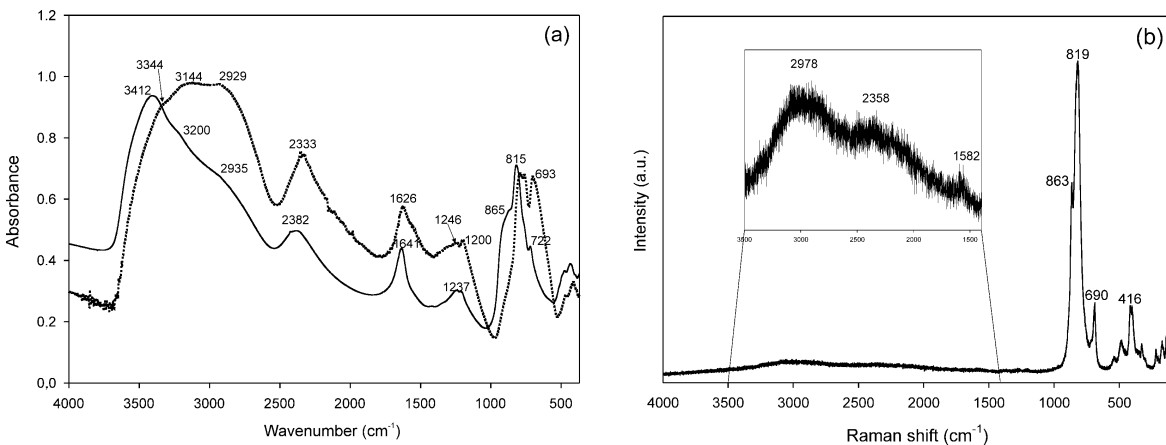

**Figure 3.** (**a**) Infrared spectra of CoAsAllu (full-line—powder spectrum; dotted line: ATR-spectrum) and (**b**) single-crystal Raman spectrum of CoAsAllu.

Correspondingly, the single-crystal Raman spectrum of CoAsAllu confirmed the presence of water and protonated arsenate groups in the spectral region between 4000–1000 cm$^{-1}$, but also suggested the possible presence of H$_3$O$^+$ anion (Figure 3b). In the region from 1000–700 cm$^{-1}$, there are the As–O stretching modes of the (HAsO$_4$)$^{2-}$ and (AsO$_4$)$^{3-}$ groups. The strongest Raman bands around 863(s) cm$^{-1}$ and 819 (vs) cm$^{-1}$ are a match to the symmetric stretching and antisymmetric stretching vibrations of the (HAsO$_4$)$^{2-}$ and (AsO$_4$)$^{3-}$ groups.

In the spectral range below 450 cm$^{-1}$, bending modes of the (HAsO$_4$)$^{2-}$ and (AsO$_4$)$^{3-}$ groups, and various lattice modes of the compound are located.

### 3.3. Magnetic Properties

The susceptibility of CoAsAllu increases with decreasing temperature until it reaches a maximum at 14.3 K. The room temperature susceptibility is 0.09 emu/mol and the product $\chi T$ = 27.5 emu K/mol as shown in inset in Figure 4. As there are 12 cobalt ions in a formula unit, an effective magnetic moment per cobalt ion is $\mu_{eff} = \sqrt{8\chi T/12} = 4.3$ $\mu_B$ [37]. This value is slightly higher from the theoretically expected value for Co(II) ions with no orbital contribution (L = 0), which amounts to 3.87 $\mu_B$, [38]. The measured effective magnetic moment of 4.3 $\mu_B$ is in a range of typically measured values for Co(II) ions with a quantum number of total electronic spin angular momentum S = 3/2 and a small but non-zero quantum number of total orbital angular momentum L [39].

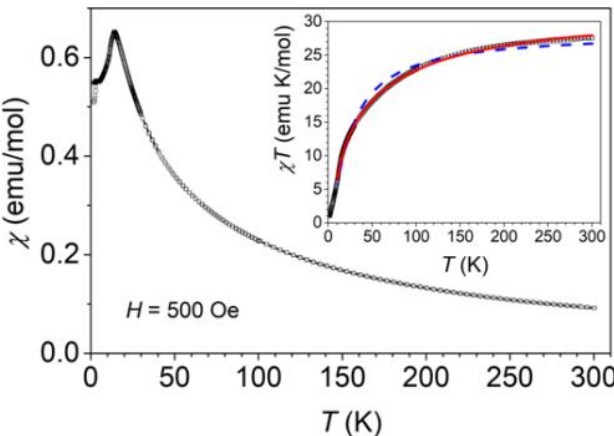

**Figure 4.** Temperature dependent susceptibility of $Co_3(AsO_4)_{0.5}(HAsO_4)_2(H_2AsO_4)_{0.5} \cdot (H_2O)_{0.5}$ and the product $\chi T$ versus temperature (inset). Blue dashed line is a fit with Equation (1), red full line is a fit with Equation (2).

The maximum of susceptibility at 14.3 K is a clear indication of an antiferromagnetic ordering. As there is no change between zfc and fc susceptibility around that maximum (both, zfc and fc susceptibilities are shown in Figure 4), the identified antiferromagnetic behavior should be associated with a low dimensional ordering. According to the structure of the CoAsAllu with a chain-like distribution of cobalt magnetic moments, the 1–D antiferromagnetic ordering is an obvious proposal. In order to estimate the interaction parameter J in the interaction Hamiltonian $H = -2J\Sigma\mathbf{S_1S_2}$ between two neighboring cobalt ions in a chain, two approaches described in [40] have been applied.

Firstly, the magnetic data $\chi T$ vs. T were fitted by Fisher's expression for a Heisenberg chain of large spins (classical vectors), [31].

$$\chi T = 12\frac{Ng^2\mu_B^2 S(S+1)}{3k_B}\frac{1+u}{1-u} \tag{1}$$

$$u = \coth\left[\frac{2JS(S+1)}{k_BT}\right] - \left[\frac{k_BT}{2JS(S+1)}\right]$$

The prefactor 12 is used owing to 12 cobalt ions in a formula unit. Other symbols obtain their usual meanings. The spin S was set to S = 3/2 and only interaction parameter J and g-factor were left as free parameters during the fitting procedure. The best correspondence between Equation (1) and the measured data for T > 10 K was obtained with $J/k_B = -7.8$ K ($-5.4$ cm$^{-1}$) and g = 2.3. The result is presented as a dashed blue line in inset in Figure 4. A somewhat large alteration between the model and the measured data is mainly because Equation (1) does not take into account the single ion effects, comparable for example with spin–orbit coupling.

Secondly, the magnetic data was analyzed using Rueff's approach [40] by means of a phenomenological equation:

$$\chi T = A\exp\left(-\frac{E_1}{k_BT}\right) + B\exp\left(-\frac{E_2}{k_BT}\right) \tag{2}$$

where $A + B$ is the Curie constant and $E_1$, and $E_2$ denote the activation energies that correspond in this case to the spin–orbit coupling (D) and the antiferromagnetic exchange interaction (J), respectively. The outstanding agreement among the experimental data for T > 10 K and Equation (2) is shown as a filled red line in inset in Figure 4 with parameters $A + B = 31.7$ emu K/mol (from which follows $g = 2.4$), $D/k_B = 100$ K, and $J/k_B = -12.2$ K ($J = -8.5$ cm$^{-1}$). The g-factor and zero filed splitting parameter D are of the same order as the values that can be found in the literature [40,41]. The interaction parameter

$J$ obtained by Rueff's approach is a bit larger from the one considering cobalt magnetic moments as classical vectors ($J/k_B$ of $-8.5$ cm$^{-1}$ versus $-5.4$ cm$^{-1}$). We can conclude that the maximum in the measured susceptibility at 14.3 K can be attributed to the intrachain antiferromagnetic interaction between neighboring Co(II) ions with the interaction parameter of the order of $J/k_B \approx -8$ cm$^{-1}$.

Figure 5a shows the temperature dependent susceptibility in an expanded scale in a low temperature region. Below 3.5 K there is a difference between zfc and fc susceptibility. The splitting between zfc and fc susceptibility is not compatible with the magnetic interaction limited only to one dimension and must be ascribed to an interchain magnetic interaction. As the fc susceptibility decreases slightly with decreasing temperature from 3.5 K down to 2 K, we can conclude that the interchain interaction is antiferromagnetic with a Néel temperature $T_N$ = 3.5 K defined as a local maximum of the fc susceptibility. A rough estimation of the interchain exchange interaction $J'$ can be obtained by comparing the temperature of the maximum of $\chi(T)$ (14.3 K) due to the intrachain interaction with the observed $T_N$ (3.5 K) due to the interchain interaction. As the intrachain interaction was calculated as $J \approx -8$ cm$^{-1}$, the interchain can be estimated of the order of $J' \approx -2$ cm$^{-1}$.

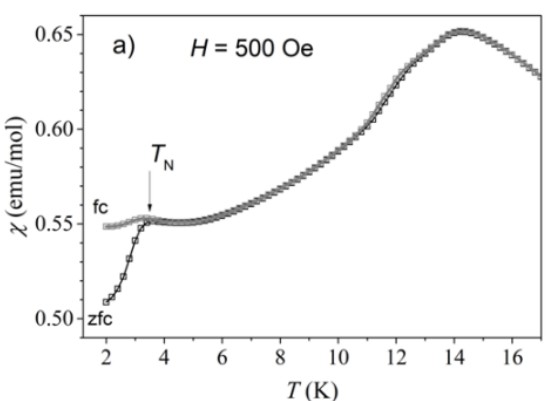
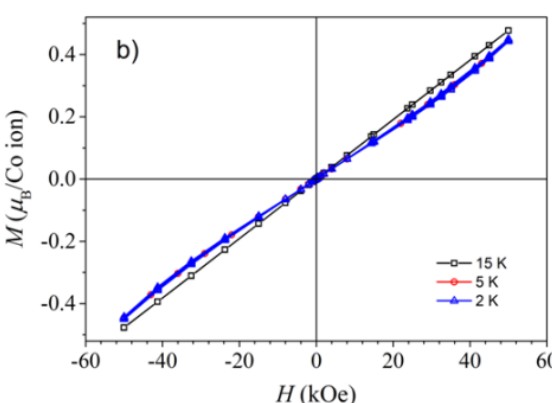

**Figure 5.** Zfc and fc susceptibility in low temperature range (**a**) and isothermal magnetization curves (**b**).

The magnetization curves $M(H)$ were measured at several temperatures. In Figure 5b, the $M(H)$ curves for three temperatures are shown. At a temperature of 15 K, slightly above the maximum of susceptibility, the $M(H)$ is linear. The $M(H)$ curves measured at 5 K and 2 K, the temperatures that are just above and below the $T_N$ at 3.4 K, coincide and their initial slope is smaller than for the $M(H)$ at 15 K. The magnetization in a maximal magnetic field that can be achieved in our measuring system, i.e., 50 kOe, is approximately 0.5 $\mu_B$/Co ion and still increases. This value is significantly lower than that predicted by Brillouin function [39] for Co(II) ions of approximately 3 $\mu_B$/Co ion. The $M(H)$ curves show no hysteresis. All these observations are in agreement with the proposed antiferromagnetic ground state. In an external magnetic field of 30 kOe the slope of $M(H)$ curves at 2 K and 5 K starts to increase. In this magnetic field the Zeeman energy of magnetic moments in external magnetic field becomes comparable to the interaction energy due to the antiferromagnetically coupled magnetic moments.

## 4. Summery

The new $Co_3(AsO_4)_{0.5+x}(HAsO_4)_{2-x}(H_2AsO_4)_{0.5+x}[(H,\square)_{0.5}(H_2O,H_3O)_{0.5}]^{2x+}$ is a part of numerous minerals and synthetic compounds with alluaudite-type structure. According to X-ray analysis, it contains a 3D framework consisting of the infinite edge-shared $CoO_6$ octahedra chains, which are interconnected by arsenate tetrahedra (Figure 2d). The statistically distributed H atoms in half-occupied sites create two $(H2_2As1O_4)^-$ and two $(As1O_4)^{3-}$ around As1, and eight $(HAs2O_4)^{2-}$ (four $(H4As2O_4)^{2-}$ and four $(H6As2O_4)^{2-}$) around two As2 in the unit cell. They carry a charge of 24–, which is balanced by the 12 Co(II) (4 Co1 and 8 Co2 at special 4*e* and general 8*f* positions, respectively). Due to the

assumed electroneutrality of the unit cell, only electroneutral species such as water can be expected in the channels. Considering 3D open-framework atoms alone, the contents of the unit cell of CoAsAllu can be written as $Co1_4Co2_8(As1O_4)_2(H_2As1O_4)_2(HAs2O_4)_8$ and including water in channel 2, it is $Co1_4Co2_8(As1O_4)_2(H_2As1O_4)_2(HAs2O_4)_8 \cdot 2H_2O$.

The infrared and Raman spectra confirmed the presence of both water molecules as well as the protonated arsenate species, but also suggested the possible presence of free $H^+$ (not detectable using X-ray diffraction) and $H_3O^+$-group. Therefore, the best way to describe the chemistry of CoAsAllu would be through formula $Co_3(AsO_4)_{0.5+x}(HAsO_4)_{2-x}$ $(H_2AsO_4)_{0.5+x}[(H,\square)_{0.5}(H_2O,H_3O)_{0.5}]^{2x+}$.

The results of the magnetic investigations confirm the expected quasi one dimensional structure of divalent cobalt ions. They are antiferromagnetically coupled with the intrachain interaction parameter of $J \approx -8$ cm$^{-1}$ and interchain parameter of $J' \approx -2$ cm$^{-1}$ that become effective below the Néel temperature of 3.4 K.

The systematics of the alluaudite-type compounds has repeatedly been reported [10–12,42]. Newlym Hatert [10] proposed the new nomenclature scheme for the minerals belonging to alluaudite-type phosphates and arsenates, which is based on the contents of the *M*1 and *M*2 octahedral sites [10] and Yakubovich [42] categorized the alluaudite-type compounds into subgroups according to the occupancy of the *A* site.

In unprotonated water bearing alluaudite-type arsenates, the *A*1 and *A*2′ sites are frequently occupied but, if vacancies occur, they are favorably located on *A*2′, as observed in yazganite, $NaMgFe^{3+}{}_2(AsO_4)_3 \cdot H_2O$, keyite, $(\square_{0.5}Cu_{0.5})CuCdZn_2(AsO_4)_3 \cdot H_2O$ and erikapohlite, $(\square_{0.5}Cu_{0.5})CuCaZn_2(AsO_4)_3 \cdot H_2O$ [10,12], where $\square$ is a vacant position. The resulting free space is filled by water molecules in channel 2 of these minerals. Accordingly, the half occupancy of *A*2′ is enough to house one water molecule. In protonated arsenates, channel 1 is occupied by the H atoms of the $[AsO_3(OH)]$ and $[AsO_2(OH)_2]$ groups, thus clarifying the presence of vacancies on the *A*1 sites of o'danielite, $Na\square ZnZn_2[AsO_4][AsO_3(OH)]_2$, canutite, $Na\square MnMn_2[AsO_4][AsO_3(OH)]_2$, magnesiocanutite, $Na\square MnMg_2[AsO_4][AsO_3(OH)]_2$ and camanchacaite, $Na\square CaMg_2[AsO_4]_2[AsO_2(OH)_2]$.

In CoAsAllu, we observe for the first time the mutual occupancy of channel 1 by H atoms from the $[AsO_3(OH)]$ group and channel 2 with water molecule and the H atom from the $[AsO_3(OH)]$ group. If we take into account the proposed classification based on the *A*-sites occupancy [42], CoAsAll represents a previously undocumented subcategory.

Due to the extremely flexible nature of the open-framework architecture in the alluaudite-type structure, we expect many further synthetic analogues of the alluaudite-type compounds with unique physical properties.

**Author Contributions:** Conceptualization, T.Đ., Z.J. and L.K.; methodology, T.Đ., Z.J. and M.J.; formal analysis, T.Đ., Z.J. and M.J.; investigation, T.Đ., L.K., Z.J. and M.J.; resources, T.Đ.; writing—original draft preparation, T.Đ.; writing—review and editing, T.Đ., L.K., Z.J. and M.J.; visualization, T.Đ., L.K. and Z.J.; project administration, T.Đ. and Z.J.; funding acquisition, T.Đ. and Z.J. All authors have read and agreed to the published version of the manuscript.

**Funding:** This work was financially supported by the Slovenian Research Agency (Grant No. P2-0348), by bilateral project Slovenia–Austria (Grant No. BI—AT/16-17-005 and SI 05/2016), by Open Access Funding by the University of Vienna and the Ministry of Education, Science and Technological Development of the Republic of Serbia (Grant III45007).

**Acknowledgments:** The authors are grateful to two anonymous reviewers and the Special issue editors for their useful comments and suggestions. We thank Uwe Kolitsch for assisting during the SEM-EDS analyses and Eugen Libowitzky for assisting during measurements of the infrared spectra. We appreciate the help of Kateřina Vejvodová in improving our English.

**Conflicts of Interest:** The authors declare no conflict of interest.

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
