# Peer review of "Water in the Alluaudite Type-Compounds: Synthesis, Crystal Structure and Magnetic Properties of Co3(AsO4)0.5+x(HAsO4)2−x(H2AsO4)0.5+x[(H,□)0.5(H2O,H3O)0.5]2x+"

_minerals, doi:10.3390/min11121372_

Round 1

Reviewer 1 Report

Crystal structure: The As-O bond length of 1.69 A is typical for neutral arsenate group AsO4. The lengths of As-OH bonds are typically > 1.73 A, but such bond elongation is not observed. This should be explained taking into account that the IR spectrum shows possible presence of isolated H+ cations (see below).

Moreover, in line 58 it is written that "monovalent cations may be replaced by protons" (and corresponding references are given). Thus, the formula of the studied compound should be rewritten taking into account this reasoning.

Line 290: the band at 3412 cm-1 does not correspond to protonated AsO4 groups (wavenumbers of AsO-H stretching vibrations are below 3200 cm-1). The band at 1237 cm-1 should be assigned (it may correspond to As-O-H bending vibrations or vibrations of isolated H+ cation formed as a result of partial dissociation of acid arsenate groups).

Figure 3: The ratio of the IR bands at 3412 and 815 cm-1 corresponds to high-hydrous acid arsenates with H2O:As > 3 (see e.g. numerous examples in the book Chukanov N.V., Chervonnyi A.D. Infrared Spectroscopy of Minerals and Related Compounds. Springer, 2016. DOI: 10.1007/978-3-319-25349-7). Thus, the bands at 3412 and 1641 cm-1 are mainly due to water molecules adsorbed by the KBr pallet and cannot be used as an indication of the presence of H2O molecules in the sample.

General comment: a more detailed discussion involving comparison of the compound studied with other synthetic alluaudite-related acid arsenates (and maybe phosphates) would be useful.

Author Response

Dear reviewer,

Many thanks for your useful suggestions concerning the infrared spectroscopy part. We took a closer look at the chapter “Acid OH groups in minerals” from the book “Infrared Spectroscopy of Minerals and Related Compounds” by Chukanov and Chervonnyi and have cited it in the manuscript. We agree with the possibility of the presence of the H+ cation, but on the basis of the IR-spectra recorded from a sample prepared as a KBr palett, we cannot exclude the possibility of the presence of the HAsO4 groups and the free water molecule. Because of that, we have recorded another IR-spectrum of the CoAsAllu using the ATR sample preparation technique. The results showed the unambiguous presence of the water molecule. Consequently, we have updated the Figure 3a, the methodology part concernig IR-spectroscopy and the dicussion on the IR-spectra.

Such as the other monovalent cations in the alluaudite structure-type, if the cation H+ exists, it should be placed in the channel together with distorted water molecule, so if the A-position in CoAsAllu is filled with water molecule and H+ the formula would be written as (H2O)0.5(H+)0.5Co2+3[(AsO4)0.75(H2AsO4)0.25]2.5–[(HAsO4)2]4–. After the consultation with Prof. Eugen Libowitzky, an expert on infrred spectroscopy and the water in minerals, we still agreed not to change the formula of the CoAsAllu, because the scenario wit acidic HAsO4 group is more probable.

Concerning the suggestion to include the detailed discussion involving comparison of the compounds studied with other synthetic alluaudite-related acid arsenates (and maybe phosphates):

In our previous works on the alluaudite type-arsenates (Stojanović et al. 2012, Đorđević et al 2015 and Đorđević et al 2017), where we have structurally and spectroscopically described five new acidic arsenates belonging to the alluaudite type-structure [Cd1.16Zn0.34(AsO4)1.5(HAsO4)(H2AsO4)0.5, Cd0.74Mg2.76(AsO4)1.5(HAsO4)(H2AsO4)0.5, NaMg3(AsO4)(AsO3OH)2, NaZn3(AsO4)(AsO3OH)2 and Cd0.75Co2.75(H0.5AsO4)2(HAsO4)] we have already done such kind of the comparison. We did not want repeat it here. In this contribution, besides the synthesis procedure and the detailed crystal structure of the title compound the focus was on the magnetic properties of the CoAsAllu.

Sincerely,

Tamara Đorđević

Reviewer 2 Report

The study by Đorđević and co-workers describes the synthesis and characterization of a new, Co-containing member of the alluaudite family of minerals. I find the manuscript scientifically sound and therefore recommend publication in Minerals.

Author Response

Dear reviewer,

We appreciate very much your recommendation for the publication.

In the behalf of all the authors,

Tamara Đorđević

Round 2

Reviewer 1 Report

I have only one but significant comment.

The mean As-O bond length of 1.69 A is typical for the neutral arsenate group AsO4. The lengths of As-OH bonds are typically > 1.73 A, but such bond elongation is not observed, and this is the most important query. The IR spectrum shows possible presence of isolated H+ cations or hydronium cations (H3O). In the latter case, strong Raman bands should be observed in the range of 1400-3000 cm-1. Unfortunately, Raman spectrum in this range is not provided.

Therefore I strongly recommend to write the formula in a more general form, e.g. as follows:

Co3[(AsO4,HAsO4)][(HAsO4,AsO4)](H,H3O,H2O)x

Author Response

Dear reviewer,

Many thanks for the further advices.

Acoording to Mähler et al. (2013), the mean As–O bond length is closer to 1.68 Å then to 1.69 Å. That the As–OH bond lengths are typically > 1.73 Å is true just for the HAsO42- anions. However, the As–O bond distances in H2AsO4- anions are shorter and varies between 1.69 Å and 1.72 Å (Mähler et al., 2013). Furthermore, for the H2AsO4- anion are typical two shorter and two longer As–O distances, which is the case for the CoAsAllu. Therefore, we do not see any reason to exclude H2AsO4- anion from the formula.

In addition, we changed Fig. 3 again and presented a single crystal Raman spectrum of CoAsAllu in the region between 1400-3500 cm–1, where the OH-stretches are clearly visible, so it is possible that the part of the water is substitued by isolated H+ and H3O+ cations. Therefore, we  agreed to write the formula of CoAsAllu as Co3(AsO4)0.5+x(HAsO4)2-x(H2AsO4)0.5+x[(H,□)0.5(H2O,H3O)0.5]x+.

The formula you proposed is not electro-neutral and it appears that we do not know how many oxygens we have in the channel, which is not true.

We believe that the new formula, Co3(AsO4)0.5+x(HAsO4)2-x(H2AsO4)0.5+x[(H,□)0.5(H2O,H3O)0.5]x+, would decribe CoAsAllu the best. Hope you will share our opinion.

Sincerely,

Tamara Đorđević

Refernce: Mähler, J., Persson, I., Herbert, R.B. (2013): Hydration of  arsenic oxyacid species. DaltonTransactions, 42, 1364–1377.